# Effect of Rice Husk Ash on the Properties of Alkali-Activated Slag Pastes: Shrinkage, Hydration and Mechanical Property

**DOI:** 10.3390/ma16083148

**Published:** 2023-04-17

**Authors:** Bo Tian, Xiangguo Li, Yang Lv, Jinsheng Xu, Weinan Ma, Chenhao He, Yang Chen, Shouwei Jian, Weizhen Wang, Cheng Zhang, Kai Wu

**Affiliations:** 1State Key Laboratory of Silicate Materials for Architectures, Wuhan University of Technology, Wuhan 430070, China; tianbo@whut.edu.cn (B.T.);; 2Key Laboratory of Advanced Civil Engineering Materials of Ministry of Education, Tongji University, Shanghai 201804, China; wukai@tongji.edu.cn

**Keywords:** alkali-activated slag, rice husk ash, hydration, shrinkage mitigation, silica fume

## Abstract

In this paper, rice husk ash (RHA) with different average pore diameters and specific surface areas was used to replace 10% slag in the preparation of alkali-activated slag (AAS) pastes. The effect of RHA addition on the shrinkage, hydration, and strength of AAS pastes was studied. The results show that RHA with a porous structure will pre-absorb part of the mixing water during paste preparation, resulting in a decrease in the fluidity of AAS pastes by 5–20 mm. RHA has a significant inhibitory effect on the shrinkage of AAS pastes. The autogenous shrinkage of AAS pastes decreases by 18–55% at 7 days, and the drying shrinkage decreases by 7–18% at 28 days. This shrinkage reduction effect weakens with the decrease in RHA particle size. RHA has no obvious effect on the type of hydration products of AAS pastes, whereas RHA after proper grinding treatment can significantly improve the hydration degree. Therefore, more hydration products are generated and fills the internal pores of the pastes, which significantly improves the mechanical properties of the AAS pastes. The 28 day compressive strength of sample R10M30 (the content of RHA is 10%, RHA milling time is 30 min) is 13 MPa higher than that of blank sample.

## 1. Introduction

At present, the issue of global carbon emission pollution has received extensive attention from countries all over the world. According to statistics, every ton of cement produced will emit about 1 ton of CO_2_ [1,2,3] into the external environment. Therefore, so as to reduce the carbon emission pollution caused by cement production, it is urgent to explore low-carbon cementitious material to replace ordinary Portland cement. Alkali-activated slag (AAS) first appeared in the 1930s, and AAS has broad potential applications [4,5]. Compared with conventional ordinary Portland cement, AAS binder has excellent properties such as low hydration heat release, low carbon emission [6], high early strength [7,8], and good durability [9,10,11]. However, due to the restrictive factors such as high shrinkage [12,13,14], short setting time [15,16], and later strength recession [17,18], AAS binder has not been widely used in practical engineering. Therefore, it is crucial to adopt appropriate materials and methods to improve the volume stability of AAS materials.

Recently, researchers have tried to prepare bio-ash obtained by burning rice husks, corn cob, peanut husks, and other agricultural solid wastes as active admixtures in the preparation of AAS pastes [19,20,21]. As one of the most common agricultural wastes, rice husks are produced in large quantities worldwide every year. According to statistics, the amount of rice husks accounts for about 22% [22,23] of the total rice production. Previous research has shown that rice husk ash(RHA) [24,25] with high amorphous silicon content can be prepared at an appropriate temperature. The prepared RHA has a loose and porous structure inside, large internal specific surface area [26,27], and high pozzolanic activity [28,29], thus it has great potential to replace cementitious materials. Zhu et al. [30] studied the influence of RHA on the microstructure of AAS pastes, and found that when the RHA content was less than 20%, it can promote the compressive strength of AAS pastes. However, when its content exceeds 20%, it has a negative effect on the mechanical property of AAS pastes. Vo et al. [31] studied the effect of MgO on the macro and micro properties of AAS and alkali-activated slag/RHA (AASR) pastes and found that the working performance of AASR pastes deteriorated with the increase of RHA content, and adding 7.5% MgO can improve the mechanical property and thermal conductivity of the AASR system. Alomayri et al. [32] studied the effect of RHA content and curing system on the properties of AAS pastes. The results show that when the RHA content is 10%, it has the best effect on the comprehensive performance improvement of AAS pastes. However, the research on the effect of RHA on the properties of AAS pastes is limited to mechanical and durability properties, and the research concerning the effect of RHA on the shrinkage and hydration properties of AAS pastes is limited. Therefore, the study of the effect of RHA on the shrinkage and hydration properties of AAS pastes has great significance.

In this paper, the influence of RHA on the working performance of AAS pastes was analyzed by means of setting time and fluidity tests. The effects of RHA on the hydration of AAS pastes were analyzed by isothermal calorimetry, XRD, and nitrogen adsorption testing techniques. The influence of RHA on the change of the internal free water signal during the hydration process was analyzed by low-field nuclear magnetic resonance testing technology. By analyzing the effects of RHA on the chemical shrinkage, autogenous shrinkage, and drying shrinkage of AAS pastes, the shrinkage inhibition mechanism on AAS pastes was explored.

## 2. Materials and Methods

### 2.1. Materials

Slag and silica fume (SF) were used in the experiment. The slag and SF are produced by SinoCem Intelligence Technology Co., Ltd. (Wuhan, China). The slag is V-800 superfine slag. The density of slag and SF was 2.81 g/cm^3^ and 2.23 g/cm^3^. RHA was produced in a muffle furnace (combustion temperature 600 °C, heating rate 10 °C/min, holding 1 h). Afterwards, the RHA was ground for 15 min and 30 min, in order to gain ash samples with different average pore diameters and specific surface areas and mesoporous structures as well. Slag, SF, and RHA tested by X-ray fluorescence spectrometry was provided in Table 1. The RHA and SF were mainly consisted of SiO_2_. The microstructures of RHA as obtained determined by SEM (QUANTA FEG450, FEI Co, Hillsboro, OR, USA) is shown in Figure 1. RHA is porous and the shape is irregular. The PSD of slag and SF is shown in Figure 2a,b. The median particle size of the slag and SF are 4.7 μm and 1 μm, respectively. The particle size distribution of RHA before and after grinding is shown in Figure 2c. For example, R-M15 means rice husk ash is ground for 15 min. Their mean particle sizes were 43.63 µm (R-M0), 17.31 µm (R-M15), and 8.57 µm (R-M30). Their average pore diameter and specific surface area were 16.6 nm (R-M0), 14.8 nm (R-M15), 10.3 nm (R-M30), and 65.6 m^2^/kg (R-M0), 55.6 m^2^/kg (R-M15), 46.8 m^2^/kg (R-M30), respectively.

The modulus of water glass (Na_2_O = 8.3 wt.%, SiO_2_ = 26.5 wt.% and H_2_O = 65.2 wt.%) is 3.3. Modulus refers to the molecular ratio (or molar ratio) of silicon oxide to alkali metal oxide in water glass. Sodium hydroxide (NaOH, 99% purity) was adopted. Naphthalene-based superplasticizer (NSP) was used to adjust fluidity. NSP is yellowish solid.

### 2.2. Sample Preparation

As shown in Table 2, the water–binder ratio (W/B) was 0.5, the sodium silicate modulus (M) was 1.4, and the alkali equivalent (N) was 8%. The incorporation of RHA was 10% of the total amount of cementitious material. SF10 indicates that SF was added to the sample, replacing the slag mass fraction by 10%. The dosage of NSP is 1% of the mass of the slag. Mixture proportions of the five groups are listed in Table 2. The preparation method of the specimen was as follows: the alkali activators for the experiment were prepared at a room temperature of 20 ± 1 °C and remained still for 1 day. In the process of slurry preparation, RHA or SF was first added to the slag by dry stirring for 2 min, and then the activator, NSP, and water were added to the slag after the mixture was evenly mixed. In the mixing machine, it was stirred slowly for 2 min at low speed (140 ± 5 r/min), stopped for 15 s, and then stirred at high speed (285 ± 5 r/min) for 2 min. Then the stirred pastes were poured into the 40 × 40 × 40 mm^3^ mold, and the mold was wrapped tightly with plastic film to prevent the pastes from contacting the outside environment. The molds were placed in a curing room at a stable temperature of 20 °C and relative humidity above 95% for curing. The specimens were demolded after 24 h. After demolding, the specimens were tightly wrapped with plastic film again. At the test age, the compressive strength was tested.

### 2.3. Testing Methods

#### 2.3.1. Fluidity and Setting Time

According to GB/T 8077-2012 [33], the fluidity of AAS slurry was measured. The test board was cleaned and leveled with a cloth. The pastes filling amount was kept flush with the upper mouth of the cone mold. After the pastes flowed freely for 30 s, the diameter of the pastes was measured in multiple directions with a ruler, and the average value was taken. Each group was measured three times and averaged. According to GB/T 1346-2011 [34], the initial setting time and final setting time were measured. Initial setting time: the specimen was tested for the first time when water was added for 30 min. The round mold was placed under the test needle so that the test needle was in contact with the surface of the c AAS slurry. The test needle sunk into the AAS slurry and observed until the test needle stop sinking, when the test needle sinks to 4 ± 1 mm away from the bottom plate, that is, the AAS slurry reaches the initial setting. Final setting time: At the end of the initial setting time, the test mold was immediately removed from the glass plate by the translation method and turned 180 degrees. The approaching final setting time was measured once every 1 min. When the test needle sinks into the specimen 0.5 mm, that is, when it cannot leave traces on the specimen, the AAS slurry reaches the final setting time.

#### 2.3.2. Chemical Shrinkage

In this study, the absolute volume method referring to ASTM C1608-17 [35] was used during hardening process. After mixing, the sample of AAS pastes is quickly weighed and 100 g was placed into a wide-mouth bottle. The bottle was filled with water and plugged with a rubber plug with a graduated tube. To prevent moisture from evaporating, liquid paraffin was used to seal all air contact areas. Readings were taken at regular intervals. The average value of chemical shrinkage value was taken from three groups of test data of each group of samples.

#### 2.3.3. Autogenous Shrinkage

Autogenous shrinkage was tested by the modified corrugated tube method according to ASTM C1698-2009(2014) [36]. The pastes were poured into the corrugated tubes. The lateral faces of the samples have edge-studs preinserted to measure the length modification, and the samples were kept at a constant temperature of 23 ± 1 °C during the automatic measurement process. Three parallel samples were tested and averaged to calculate the autogenous shrinkage value for each mixture.

#### 2.3.4. Drying Shrinkage

The pastes were poured in to a triple stainless steel mold of 25 × 25 × 280 mm^3^. According to standard GB/T50082-2009 [37], the paste samples were wrapped tightly with plastic film and then cured at 20 ± 1 °C for 24 h. After demolding, the surface water of samples was dried, and the initial length was measured. Consequently, the length of the samples was regularly recorded with a length gauge.

#### 2.3.5. Isothermal Calorimetry

An 8-channel TAM isothermal calorimeter was used, and the temperature varied from 5 °C to 90 °C. Each 20 mL glass bottle was filled with fresh pastes with mass of 5 g. The measurement temperature was 20 °C. The heat release of pastes at the first 3 days was obtained which was used to characterize the hydration reaction rate and the cumulative heat release of hydration.

#### 2.3.6. XRD

The hydration products were analyzed by means of an X-ray diffractometer (XRD, D/Max-RB). The instrument parameters: radiation source was copper target (Cu, Kα), ceramic optical tube of special size, acceleration voltage was 40 kV. The scanning angle (2θ) was between 10° and 80°, scanning speed was 4°/min, step size was 0.02.

#### 2.3.7. BET

The pore structure of the samples was analyzed by nitrogen adsorption instrument (ASAP 2460, Micromeritics, Norcross, GA, USA). Instrument test parameters were as follows: relative pressure P0/P was between 0.05 and 0.99. About 2–3 g of hardened AAS pastes, 1–2 mm thick, were dried in a vacuum oven. The vacuum oven temperature was 50 °C. Nitrogen adsorption data were processed by the Barrett–Joyner–Halenda (BJH) method for pore structure analysis of samples.

#### 2.3.8. Solid State NMR

The ^29^Si spectrum of the samples were measured by solid state ^29^Si MAS NMR (Advance III 400 MHz, Bruker, Germany). Instrument test parameters: acquisition frequency of 79.5 MHz. The sample preparation method is the same as XRD. About 50 mg of the sample is put into a 7 mm zirconia rotor. The temperature was a constant 20 °C. ^29^Si NMR instrument test parameters: acquisition frequency of 79.5 MHz, rotation speed of 5 kHz, delay time of 10 s. XPS peak fit software version 4.1 was used for data fitting.

#### 2.3.9. ^1^H Low Field NMR

The hydrogen proton relaxation signal and the change of the internal free water was detected by a ^1^H low field NMR instrument (MesoMR12–060 H-1, Niumai Analytical Instrument Co., Ltd., Suzhou, China). The magnetic field intensity was 0.5 T and the π/2 and π pulse lengths were 7.5 μs and 15 μs, respectively. The T_2_ relaxation time was inverted to characterize the free water signal in the samples. For the sake of increase in the signal-to-noise ratio (SNR), the samples were scanned repeatedly 64 times. The monitoring temperature of the samples was maintained at 20 ± 1 °C, and the test time was from 1 h after the completion of pastes mixing with water to 7 days.

#### 2.3.10. Compressive Strength

According to GB/T 17671-1999 [38], automatic cement flexural and compressive testing machine (TYE-3000, Jianyi Instrument & Machinery Co., Ltd., Changzhou, China) was used to test the compressive strength of AAS samples at the curing ages 1 day, 7 days, and 28 days.

## 3. Results

### 3.1. Fluidity

The fluidity of AAS pastes with different RHA and SF contents are shown in Figure 3. It can be seen that the fluidity of SF10 group is the largest, which is 205 mm. Additionally, the incorporation of SF can increase the fluidity of pastes. The fluidity of the R10M0, R10M15, and R10M30 groups was 180 mm, 190 mm, and 195 mm, which were 20 mm, 10 mm, and 5 mm lower than Ref, respectively. It can be found that the addition of RHA will reduce the fluidity of pastes to different degrees depending on the internal pores of RHA, which is different from that of SF. The reason is that RHA replaces part of the slag, and its internal porous structure will absorb part of free water. As a result, the effective water–binder ratio of cementitious material in the early stage is reduced, so the fluidity of AAS pastes is reduced.

### 3.2. Setting Time

As shown in Figure 4, the initial setting time of the SF10 is 30 min and the final setting time is 35 min. The initial setting time of R10M0, R10M15, and R10M30 is 28 min, 25 min, and 22 min, and the final setting time is 33 min, 30 min, and 27 min, respectively. Compared with Ref, the initial setting and final setting time of the three groups are prolonged by 8 min, 5 min, and 2 min. The incorporation of SF and RHA will prolong the setting time of the AAS pastes. The elongation effect caused by the incorporation of SF is stronger than RHA. The reason is that the pozzolanic activity of the slag and RHA is lower than the slag, resulting in a lower hydration rate and a longer setting time in the early stage due to dilution effect. With the decrease of the powder average pore diameter of RHA, the effect of prolonging the setting time of AAS pastes gradually decreases. The reason is that as the specific surface area of RHA decreases, the pozzolanic activity increases and the dilution effect decreases.

### 3.3. Hydration Properties

#### 3.3.1. Isothermal Calorimetry

From Figure 5a, compared with the Ref, the incorporation of SF and RHA hardly affects the number of hydration exothermic peak of the AAS pastes. Two main exothermic peaks can be noticed, corresponding to the first slag dissolution peak and the second hydration exothermic peak. The first exothermic peak occurs at 0–0.5 h, which is due to the initial dissolution reaction when the glass structure of the slag comes into contact with the alkali activator. Then a main peak appears, indicating the accelerated reaction between slag and alkali activator, namely the acceleration period. The maximum height of the second exothermic peak for Ref, SF10, R10M0, R10M15, and R10M30 is reached at around 2.8 h, 4.5 h, 4.8 h, 3.6 h, and 2.9 h, respectively. The heat flow for Ref, SF10, R10M0, R10M15 and R10M30 is 3.75 mW/g, 2.3 mW/g, 2.19 mW/g, 3.09 mW/g, and 3.36 mW/g, respectively. It is noteworthy that the incorporation of SF or RHA will reduce the peak value of the second hydration exothermic peak and overall shifts to the right. In addition, with the reduction of RHA particle size, the second peak shifted to the left and the peak gradually increased. It can also be found from Figure 5b that the cumulative heat of R10M0 is lower than Ref, and the cumulative heat of SF10, R10M15, and R10M30 is higher than Ref. From previous studies [39], it can be found that the incorporation of SF will delay the dissolution of slag and reduce the intensity of hydration reaction, resulting in the shift of the second hydration exothermic peak to the right and decrease the peak value, but as the hydration reaction proceeded, the [SiO_4_]^4−^ in the SF is gradually dissolved out and participate in the hydration reaction. Therefore, the cumulative hydration heat is higher than Ref. However, due to internal porous structure of RHA, it can absorb part of the pore solution [40]. Therefore, the hydration reaction rate will decrease. With the continuous progress of the hydration reaction, the reactive ions will enter the internal pore structure of the RHA particles and undergo further hydration reaction with the pore solution. Therefore, the cumulative exothermic heat of R10M30 and R10M15 is higher than that of Ref. In the R10M0 sample, the cumulative hydration heat release is lower than Ref, the reason may be due to the low degree of mechanical activity of R-M0.

#### 3.3.2. XRD

The XRD patterns of AAS products cured for 28 days are demonstrated in Figure 6. After curing for 28 days, the XRD peaks of AAS products scarcely changed, and the main hydration products are C-A-S-H gel and hydrotalcite-like phase as Ref. The C-A-S-H gel is mainly distributed at the position of 2θ = 29°, which is a cryptocrystalline diffraction peak with large half-peak width. The hydrotalcite-like phase is mainly distributed at 2θ = 12° and 39°, which are crystalline diffraction peaks with poor crystallinity. In addition, there are still some diffraction peaks of calcite at 2θ = 23°, 29°, 34°, and 39°, which is because the samples are carbonized when it comes into contact with air during the preparation process [41]. This indicates that the incorporation of RHA and SF has no obvious effect on the type of hydration products of AAS pastes. It is noteworthy that after the incorporation of RHA and SF, the C-A-S-H gel peaks at 2θ = 7° and 50° become more obvious. This is ascribed to that both RHA and SF have pozzolanic activity, which can boost the dissolution of slag, thereby producing more AAS products.

#### 3.3.3. Pore Structure

The pore structures of samples are analyzed cured for 7 days (Figure 7a,b) and 28 days (Figure 7c,d). From Figure 7, it is noticed that the porosity of SF10 is lower than Ref, the most probable radius also decreases. The reason is that the incorporation of SF has a refining effect on the internal pore structure of the AAS matrix. SF not only has a certain promotion effect on slag hydration. At the same time, because of its micro-filling effect, the larger pores inside the AAS samples can be filled, thereby the porosity is reduced [42,43]. Additionally, as shown in Figure 7a,b, compared with the Ref, the incorporation of RHA will increase the most probable radius of the AAS samples to different degrees. The pore size distribution curve shifts to the right accordingly. This is related to the internal porous structure of RHA. However, as can be seen from Figure 7c,d, when the curing age reaches 28 days, the porosity of samples mixed with RHA decreases, and the curve also moves to the left. The R10M30 has the smallest maximum pore size and the most densified internal pore structure. The porosity of R10M15 and R10M0 are still higher than that of Ref, and the pore size of R10M0 is still relatively large. The reason might be that the R-M0 is not fine enough, the pozzolanic activity is low, and the C-A-S-H gel generation is less. The internal pores of the samples are difficult to compact.

#### 3.3.4. Solid State NMR Analysis

The ^29^Si MAS NMR spectra of mixture of slag and RHA, slag and SF, and AAS pastes cured for 28 days are shown in Figure 8a–g, respectively. From Figure 8c–g, with the hydration and dissolution of the slag, the peak intensity of the Q0 peak decreases at about −74.6 ppm. Moreover, there are five additional peaks corresponding to different hydration products [44,45]. The Q^1^-a and Q^1^-b peaks are at around −77.1 ppm and −78.7 ppm, which mainly represent chain-end Si-O tetrahedron without specifically assigning each to a particular chemical environment. The Q^2^(1Al) peak is at −88.5 ppm, which mainly represents a middle-chain Si-O tetrahedron in which an adjacent Si in C-(A) -S-H gels is replaced by an Al [46]. Q^2^-b and Q^2^-p mainly represent the middle-chain Si-O tetrahedron, located at −84.2 ppm and −86.8 ppm, respectively [47]. As shown in Figure 8a,b the NMR curve of the slag is a peak with a distribution range of from −50 to −100 ppm, and the center of the peak is about −74.6 ppm. In addition, the incorporation of RHA and SF will increase a broad peak of raw materials compared with raw slag, and the center of the peak is about −114 ppm (Q^4^-a) and −111 ppm (Q^4^-b), respectively [25,48]. This could also explain their pozzolanic activity. It can be seen from Figure 8c–g that with the hydration of slag, the Q^4^-a peak in hydration products of RHA disappeared, whereas the peak intensity of Q^4^-b in hydration products of SF10 decreases. The reason is that the active silicon in RHA and SF is consumed by the hydration reaction.

In addition, integrated intensities of Si-O tetrahedron can reflect the degree of hydration (Rs) of the slag, and can be obtained according to the corresponding calculation formula [49], the calculation results are shown in Table 3. It can be seen from Table 3 that the Rs of SF10 is 67.9%, which increase 7.26% compared with Ref. With the increase of the average pore diameter of RHA, the integrated intensity of Q^2^(1Al) peak and Rs increase gradually. The Rs of R10M0 is 5.28% lower than Ref. The Rs of R10M15 and R10M30 are 69.31% and 69.97%, respectively. The difference between them is only 0.66%. The reason is that the activity of RHA can be significantly increased by proper milling, whereas the activity of RHA can be decreased by excessive milling, and even the activity of pozzolanic ash may even become lower. Compared with the SF10, the Rs of R10M15 and R10M30 are higher. The reason is that not only due to the effect of mechanical activation on RHA, but also related to the internal curing effect of its internal porous structure. Therefore, RHA after proper grinding is better than SF in promoting hydration reaction, being consistent with the results of isothermal calorimetry.

#### 3.3.5. ^1^H Low Field NMR

The ^1^H low field NMR technique was applied, so as to further explore the effect of RHA on the shrinkage mitigation mechanisms. The effects of RHA on T_2_ relaxation time distribution and signal intensity changes in free water inside AAS samples (see Figure 9 and Figure 10). From Figure 9, the T_2_ relaxation time is mainly distributed within 0.1–10 ms. As the hydration reaction progresses, T_2_ gradually shortens and the signal of free water peak shifts to the left. This is due to the increase of hydration degree, resulting in the consumption of free water [50,51,52]. With the decrease of the peak value of free water signal, the hydration age increases gradually. The hydration ages were 0.5 h, 1 h, 2 h, 3 h, 4 h, 5 h, 6 h, 7 h, 8 h, 9 h, 10 h, 12 h, 24 h, 48 h, 72 h, 96 h, 120 h, 144 h and 168 h.

So as to better observe the influence of RHA and SF on the internal free water consumption process of AAS sample, the free water signal was then converted into a percentage of the initial signal strength. The initial relaxation peak signal strength was set to 100% and then the free water signal strength was measured each time compared to the initial relaxation peak signal strength. From Figure 10, the free water signal ratio of the samples decreases rapidly from about 3–5 h. The change of free water signal ratio of R10M0 is the smallest. The change of free water signal ratio of R10M30 is the most significant and the samples of R10M15 and SF10 are lower than Ref. It indicates that the pozzolanic activity of R10M30 is the highest, and it promotes the hydration and dissolution of slag most obviously. In addition, the incorporation of SF and RHA has no obvious effect when the free water signal ratio starts to decrease. This may be attributed to the drastic hydration reaction of the samples caused by the high alkali equivalent of the alkaline activator. When the sodium oxide equivalent is 8%, almost no transition period is induced in the early hydration heat release process, which is consistent with the results in Figure 5.

With the progress of hydration reaction, the signal fraction of R10M15 and SF10 is lower than Ref at about 54 h and 144 h, respectively. At 168 h, the signal fraction of the SF10 decreased by 54.6%, which is still higher than Ref by 1.2%. This is because the SF has strong pozzolanic activity, thus promoting the hydration and dissolution of slag in the hydration age. The signal fraction of R10M15 decreased by 54.3%, being attributed to the fact that the porous structure inside the RHA can absorb part of the pore solution. The active silicon inside can be dissolved out, and the free water consumed by hydration reaction can be added, which has an internal curing effect. In addition, as the specific surface area of RHA decreases, the internal free water signal decreases more. The reason is that the smaller the specific surface area, the higher the pozzolanic activity of RHA, and thus the higher the content of free water consumed inside the samples.

### 3.4. Shrinkage Performance

#### 3.4.1. Chemical Shrinkage

Figure 11 shows the evolution of chemical shrinkage of AAS pastes. In the hydration process of cement, the reaction product is smaller than the volume of slurry. Take C_3_S as an example. After hydration, the volume decreases by 6.6%. The resulting volume shrinkage of concrete is called chemical shrinkage. During the first 72 h, the chemical shrinkage decreases, being consistent with the autogenous shrinkage. It can be seen from Figure 11 that during the first 14 days, the chemical shrinkage value of SF10, R10M15, and R10M30 is larger than Ref, among which R10M30 is the largest, which can reach 0.032 mL/g. It indicates that both SF and RHA after grinding treatment could promote the hydration of slag. The reason is that both SF and RHA contain a lot of active silicon, which is dissolved by pore solution, promoting the hydration reaction inside the AAS pastes. In addition, the pozzolanic activity of RHA is lower than SF [30]. However, the chemical shrinkage of the R10M15 and R10M30 are higher than SF10 after 14 days. The phenomenon can be ascribed to two reasons: On the one hand, mechanical grinding reduces the particle size of RHA and increases internal specific surface area. The reaction probability of Ca^2+^ in pore solution with active silicon in RHA is increased. Therefore, the pozzolanic activity is improved. On the other hand, there are many pore structures inside RHA particles, part of the water can be absorbed and released in the process of slag hydration, which has the effect of internal curing [53], improving the hydration degree. Compared with the R10M0, the chemical shrinkage value of the R10M15 increased 0.01 mL/g. With the further reduction of the specific surface area, the chemical shrinkage value of R10M30 was only 0.002 mL/g larger than R10M15. It indicates that grinding treatment of RHA can significantly increase its pozzolanic activity, but the rate of increase is gradually reduced.

#### 3.4.2. Autogenous Shrinkage

The incorporation of SF and RHA can reduce the early autogenous shrinkage of the samples in Figure 12. There is usually a large shrinkage before the final setting time, which usually exceeds 70% of the autogenous shrinkage value. Shrinkage at this stage is mainly caused by the volume reduction caused by cement hydration, which is mainly dominated by chemical shrinkage. Autogenous shrinkage continues to develop after final setting time. This part of shrinkage is mainly caused by the reduction of internal humidity as cement hydration continues. Early cement rapidly hydrated, causing this part of the shrinkage also developed rapidly. The shrinkage before final setting time is mainly manifested as the change of absolute volume, which is usually not considered in the autogenous shrinkage studies of AAS pastes. All the autogenous shrinkage studied in this paper starts after final setting time.

The autogenous shrinkage of the Ref sample was 9023 μm/m. Compared with the Ref, the autogenous shrinkage of SF10 sample decreased slightly by 5% to 8573 μm/m. This indicates that the incorporation of SF has no obvious effect on the early autogenous shrinkage of AAS pastes. The slight decrease in autogenous shrinkage is because the incorporation of SF makes part of the OH^−^ in the reaction system to dissolve the SiO_2_ in SF, which has a dilution effect on the whole reaction system. The hydration reaction rate is slowed down, and the overall hydration reaction is delayed [54].

At 168 h, the value of autogenous shrinkage for R10M30, R10M15, and R10M0 are 7382 μm/m, 6877 μm/m, and 4269 μm/m, which decreased by 18%, 23.8%, and 52.7%, respectively, compared with the Ref. With the increase of grinding particle size, the inhibitory effect of RHA on the early autogenous shrinkage gradually decreases. The R10M0 has the best shrinkage inhibition effect. The phenomenon can be ascribed to two reasons: (ⅰ) the internal specific surface area of raw ash is small, which leads to the reduction of the pozzolanic activity, the intensity of hydration reaction and the consumption of internal free water. (ⅱ) RHA has internal curing effect [55]. The incorporation of a larger pore structure absorbs part of the free water and releases it during the slag hydration process, which alleviates the decrease in the relative humidity inside the AAS pastes. With the increase of average pore diameter, the pozzolanic activity of RHA gradually increases, and the consumption of free water inside the sample increases. At the same time, the capillary diameter increases resulting in an increase in the internal capillary pore pressure. Therefore, the autogenous shrinkage value of R10M15 increased. At the same time, because it also contains more pore structures, it acts as an internal curing agent in the hydration process. Therefore, it also has a certain inhibitory and alleviation effect. In addition, compared with R10M15, the autogenous shrinkage of R10M30 has a slight increase, and the autogenous shrinkage alleviation effect is the worst. This shows that with the further decrease of average pore diameter, the pozzolanic activity of RHA is not significantly improved, and the effect of promoting slag hydration has little change. This may be attributable to the destruction of the porous structure inside part of it, resulting in a decrease in its internal curing effect.

#### 3.4.3. Drying Shrinkage

Figure 13 shows the results of drying shrinkage of all samples. Drying shrinkage is the shrinkage of cementitious materials due to the increase of capillary wall pressure caused by the loss of water in gels and capillary pores. Drying shrinkage is an important cause of matrix cracking and producing cracks.

Compared with the Ref, the drying shrinkage value of SF10 was slightly reduced by only 1.6%. The phenomenon can be ascribed to two reasons: (ⅰ) dilution effect. The addition of SF will cause dilution effect to the reaction system [56], and the reaction rate of slag dissolution and hydration will decrease. (ⅱ) filling effect. The particle size of SF is small, which can fill the internal pores of the samples to a certain extent [57], which can also inhibit the increase of pore pressure caused by the consumption of internal free water by hydration. However, the addition of RHA alleviates the drying shrinkage significantly, and the grinding size of RHA has a great influence on the reduction of drying shrinkage value. At 28 d, the values of drying shrinkage for R10M0, R10M15 and R10M30 are 9500 μm/m, 10,141 μm/m and 10,650 μm/m, which decrease by about 17.1%, 11.5% and 7.1% compared with Ref, respectively. It is observed that the alleviating effect of RHA on drying shrinkage decreases with the decrease of RHA grinding particle size.

Additionally, the drying shrinkage of R10M0 decreases most obviously. The drying shrinkage of the AAS pastes is mainly provided by the autogenous shrinkage in the first 7 d, and the 7 d–28 d is mainly related to the volatilization of the internal water into the dry environment. Therefore, the reason why the raw ash reduces the drying shrinkage of the sample most obviously is precisely because it has a great alleviating and inhibiting effect on the early autogenous shrinkage of the sample, which is closely related to the mitigating effect, low pozzolanic activity and internal curing effect. The pozzolanic activity of RHA increased gradually with the decrease of particle size. The effect of alleviating the early autogenous shrinkage of the sample becomes poor. As a result, the drying shrinkage values of R10M15 and R10M30 samples are larger than R10M0. This is consistent with the autogenous shrinkage result.

### 3.5. Mechanical Property

The compressive strength of the Ref is 94 MPa at 3 days, while the compressive strength of SF10 is always slightly higher (1–3 MPa) than Ref and R10M0 is lower than Ref in Figure 14. The reason is that the pozzolanic activity of R-M0 is low, which reduces the hydration degree of slag, and the internal porosity of the sample is high, resulting in the decrease of the strength. Compared with reference samples, the compressive strength loss rate is 28.7% at 3 days. As hydration of slag continues, the compressive strength loss rate decrease to 27.2% and 8.4% at 7 days and 28 days. The loss rate decreased significantly. The reason is that the porous structure inside RHA can absorb part of the free water and release it to participate in the hydration reaction at the late age of hydration, which improves the hydration degree of slag and generates more gels dense pore structures, and the compressive strength loss is gradually compensated at later age. The compressive strength of R10M15 is slightly lower than Ref by 1–2 MPa. The reason might be that the porosity of R10M15 is still higher than Ref. The R10M30 is always higher than Ref, and its compressive strength can reach 155 MPa at 28 days, 13 MPa higher than reference samples. This is mainly because the pozzolanic activity increases significantly with the decrease of RHA particle size.

## 4. Discussion 

### 4.1. Hydration Kinetics

The addition of RHA will delay the exothermic peak of hydration. The reason is that RHA will absorb some of the pore solution in the AAS pastes and reduce the hydration reaction rate. With the progress of hydration reaction, the cumulative hydration heat of R10M15 and R10M30 is higher than that of Ref. In addition, solid state ^29^Si MAS NMR and pore structure analyses also show that the hydration degree of R10M15 and R10M30 is higher than that of Ref, whereas the total porosity is the opposite. The reason is that the pozzolanic activity of RHA increases with the decrease of specific surface area, which greatly promotes the internal hydration reaction of the samples. Thus, the cumulative heat release of hydration and the total hydration degree increases. At the same time, the pores inside the samples are also filled with more hydration products, therefore the pore structure inside the matrix is relatively compact. However, the RHA pozzolanic activity of R10M0 is still low and the cumulative heat release decrease due to the reduction of slag content in the reaction system. Therefore, the internal hydration degree is also lower than Ref, and the pores in the matrix cannot be filled with hydration products, therefore, the total porosity is higher than Ref significantly. In addition, the increase of hydration product content in R10M15 is higher than R10M30. The reason is that the excessive grinding will destroy the porous structure inside RHA, and the inner curing effect will degrade. The result confirms that with proper milling treatment, RHA can not only play the role of internal curing agent, so that its shrinkage performance has been improved, but also promote the internal hydration and optimize the internal pore structure, thus improving the mechanical properties. However, the compressive strength of AAS pastes is greatly affected by the specific surface area of RHA. The compressive strength of R10M15 and R10M0 at a later hydration stage was lower than that of blank group. This is due to the low pozzolanic activity of RHA with larger particle size. Meanwhile, some of the water stored in the pores of the porous structure of RHA is not fully released.

### 4.2. Shrinkage-Mitigating Mechanism of RHA

The incorporation of RHA in R10M15 and R10M30 promotes the development of chemical shrinkage of AAS pastes at 14 days. The reason is that the particle size of rice husk ash becomes smaller after grinding, and the specific surface area increases, which is more conducive to providing active silicon for the hydration reaction, which is consistent with the NMR. At the same time, its internal porous structure can also absorb part of the pore solution and further increases the hydration degree in the later age of hydration. This promotion is even stronger than SF. RHA can also alleviate the early autogenous shrinkage of AAS pastes, and the alleviating effect of raw ash is the most significant. The reason is that the pozzolanic activity of raw ash is poor, and it has little effect on the hydration reaction. And just like the incorporation of SF, there is a dilution effect on the whole reaction system due to the reduction of slag content. Another reason might be that the pore structure inside the raw ash is relatively intact and has a good curing effect, which can alleviate the reduction of the relative humidity inside the samples, being consistent with the results of ^1^H NMR. With the decrease of the average pore diameter of RHA, the pozzolanic activity of RHA increases, which can promote the hydration reaction inside the samples. However, the internal pore structure of RHA is destroyed by milling, and its effect on alleviating the relative humidity reduction is weakened. As a result, the autogenous shrinkage of R10M30 and R10M15 samples is increased, but still lower than SF10 and Ref samples.

After the hydration of 28 days, RHA can reduce the drying shrinkage significantly. The reason is that RHA can inhibit the autogenous shrinkage of AAS pastes. Another reason might be that RHA can promote the hydration of AAS pastes, and generate more hydration products to fill the pores inside the matrix, which can reduce the degree of water loss of AAS pastes in dry environment, so that the drying shrinkage of AAS pastes is reduced by about 5–10% compared with SF10. The result confirms that after proper milling treatment, RHA will perform high pozzolanic activity and internal curing effect, which is stronger than SF10 in promoting hydration and reducing shrinkage of AAS pastes.

## 5. Conclusions

In this paper, the shrinkage-mitigating mechanism of RHA and hydration kinetics of AAS pastes containing RHA was studied, as well as the strength development. Based on the research above, the main conclusions can be drawn as follows:The incorporation of RHA will cause the fluidity loss of AAS pastes, and the loss degree decreases with the decrease of RHA average pore diameter. Fluidity loss is induced by water uptake of internal pores. In addition, RHA can prolong the setting time of AAS pastes, and the setting time is shortened with the decrease of the average pore diameter;RHA retards the arrival time of hydration reaction peak of AAS pastes, while this retarding effect is mitigated with the decreased average pore diameter of RHA. However, RHA increases the cumulative hydration heat of AAS pastes. In addition, the hydration degree of AAS pastes is also promoted by adding RHA;The incorporation of RHA increases the chemical shrinkage of AAS pastes. However, RHA can alleviate autogenous shrinkage at 7 days and drying shrinkage at 28 days of AAS pastes. The alleviating effect deteriorated with the decrease of the average pore diameter of RHA;The incorporation of RHA coarsens the early pore structure of AAS pastes and increased the total porosity of samples. However, with the progress of hydration reaction, the internal pore structure is refined at later age, and the total porosity decreases;The incorporation of RHA can significantly enhance the compressive strength of AAS pastes at later age. For example, by adding 10% RHA with a D50 of 8.3 μm, the compressive strength of AAS pastes can reach 155 MPa at 28 days, increasing by 13 MPa compared with that of AAS pastes without RHA.

## Figures and Tables

**Figure 1 materials-16-03148-f001:**
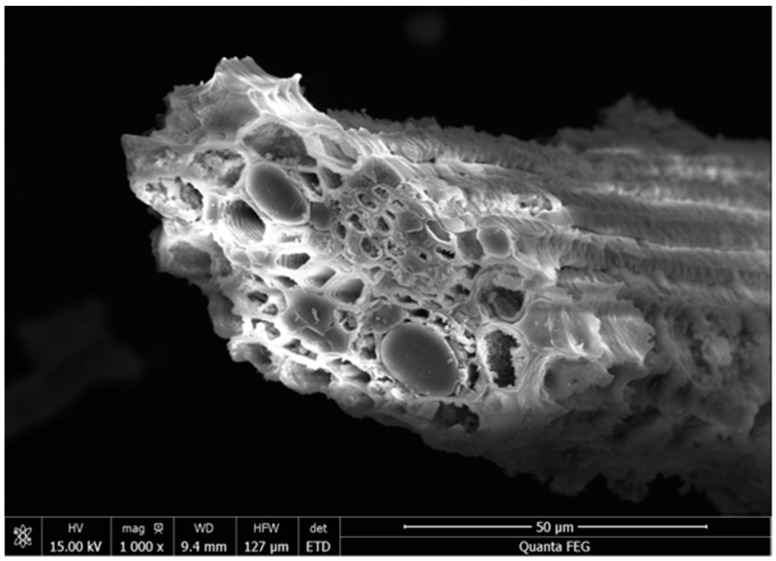
SEM image of RHA.

**Figure 2 materials-16-03148-f002:**
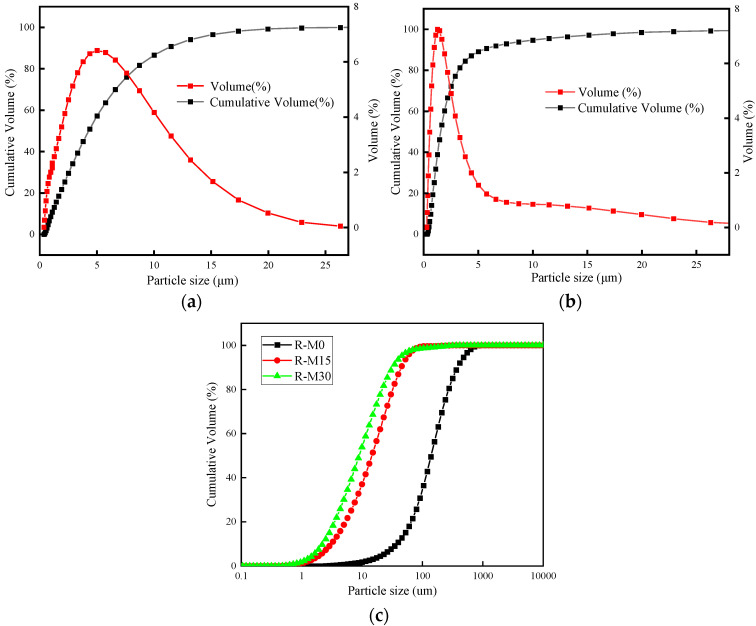
Particle size distribution of materials. (**a**) Slag, (**b**) SF, (**c**) RHA.

**Figure 3 materials-16-03148-f003:**
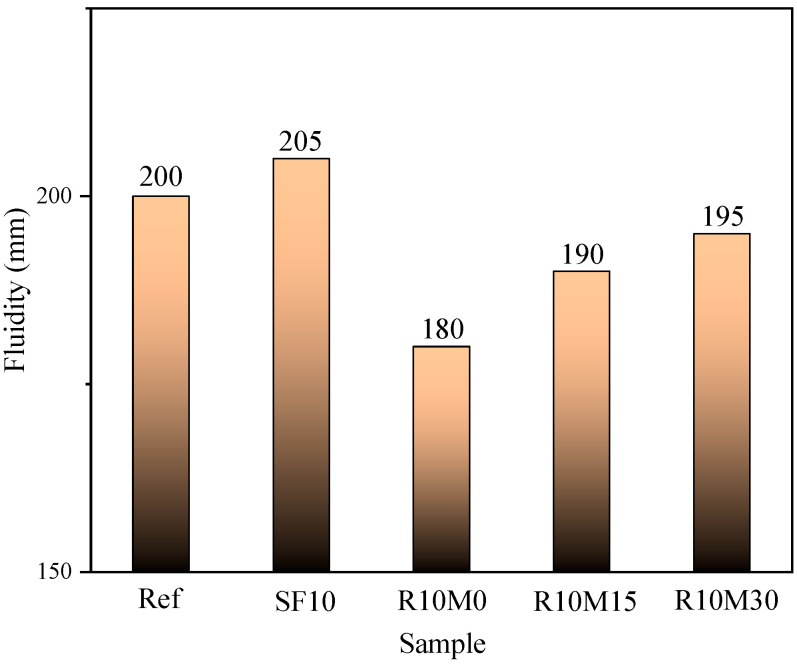
Fluidity of AAS pastes.

**Figure 4 materials-16-03148-f004:**
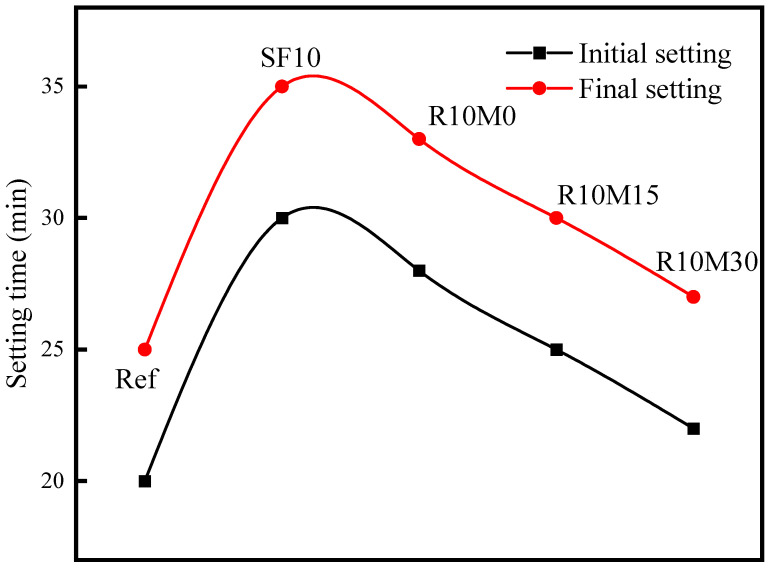
Setting time of AAS pastes.

**Figure 5 materials-16-03148-f005:**
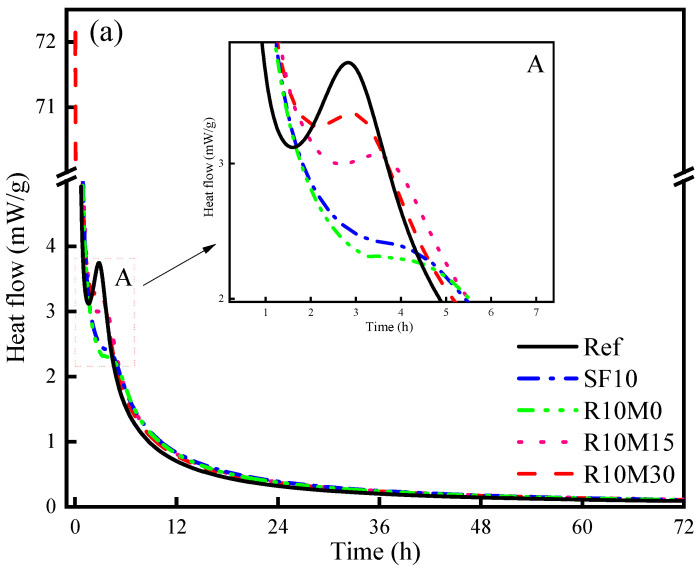
Heat flow (**a**) and cumulative heat (**b**) of AAS pastes.

**Figure 6 materials-16-03148-f006:**
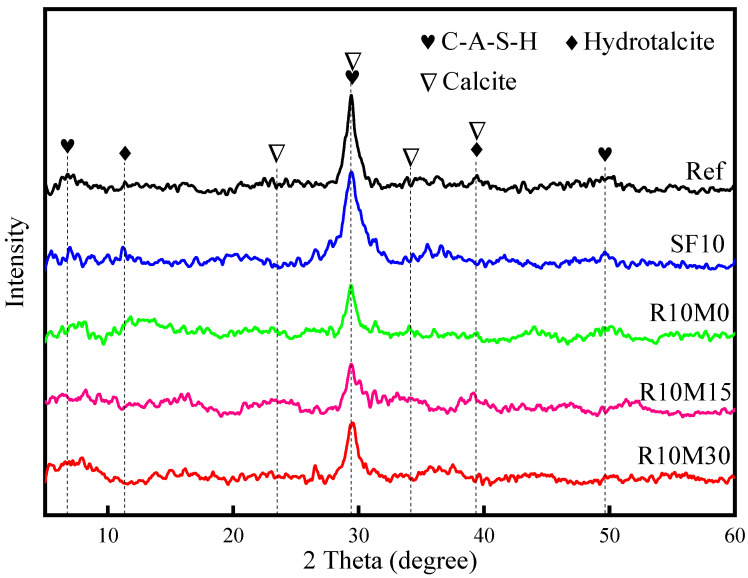
XRD patterns of hydration products of AAS samples.

**Figure 7 materials-16-03148-f007:**
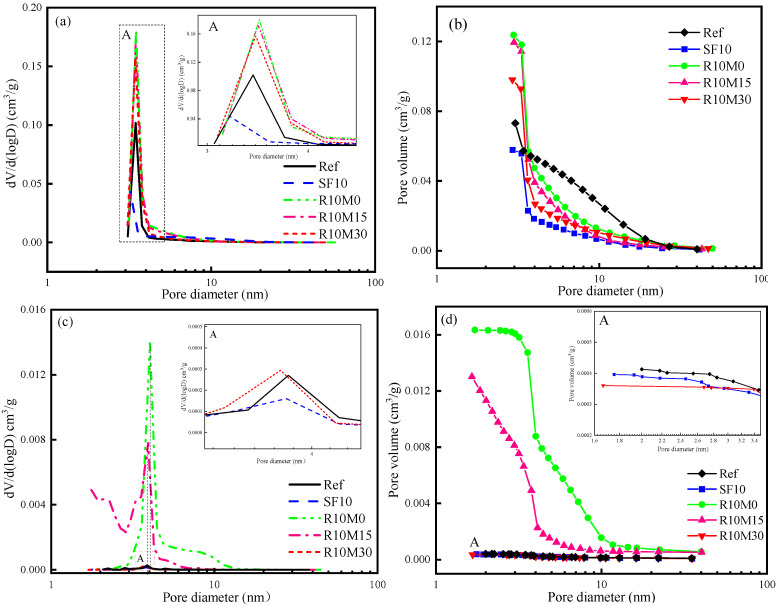
Pore size distribution and cumulative pore volume of AAS samples at 7 days (**a**,**b**) and 28 days (**c**,**d**).

**Figure 8 materials-16-03148-f008:**
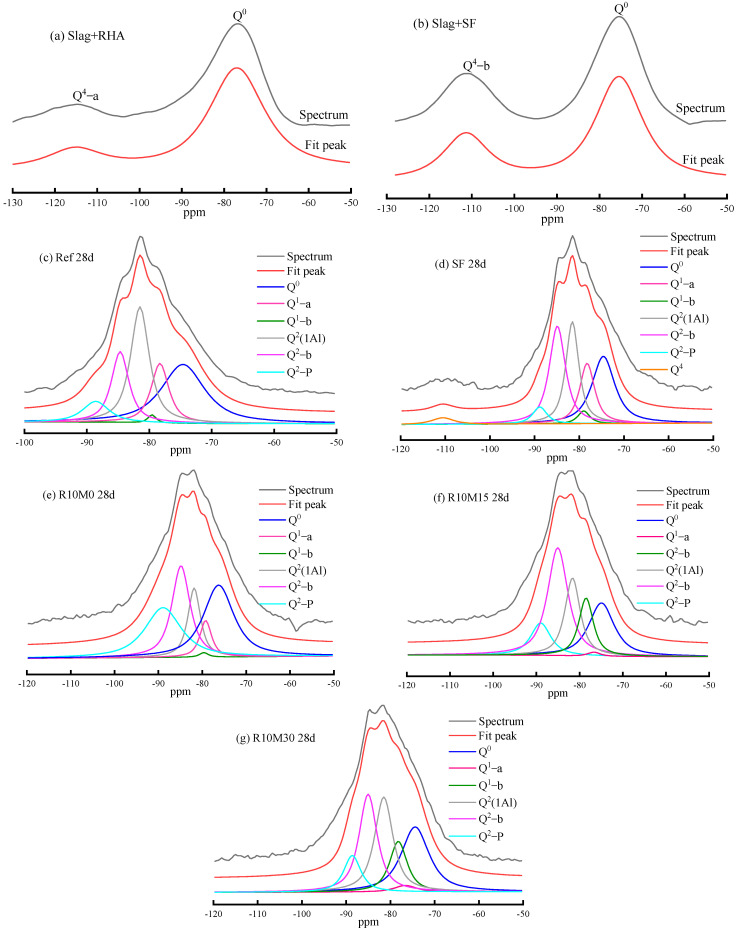
The ^29^Si MAS NMR spectra.

**Figure 9 materials-16-03148-f009:**
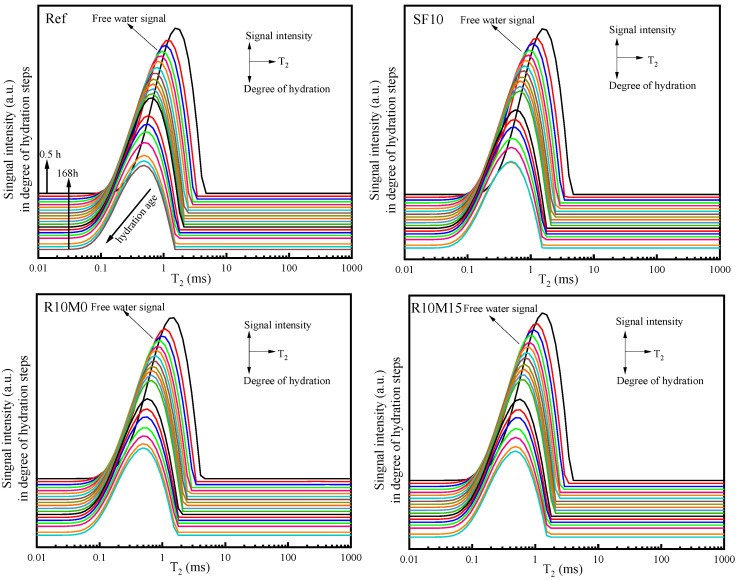
T_2_ relaxation distributions of AAS pastes.

**Figure 10 materials-16-03148-f010:**
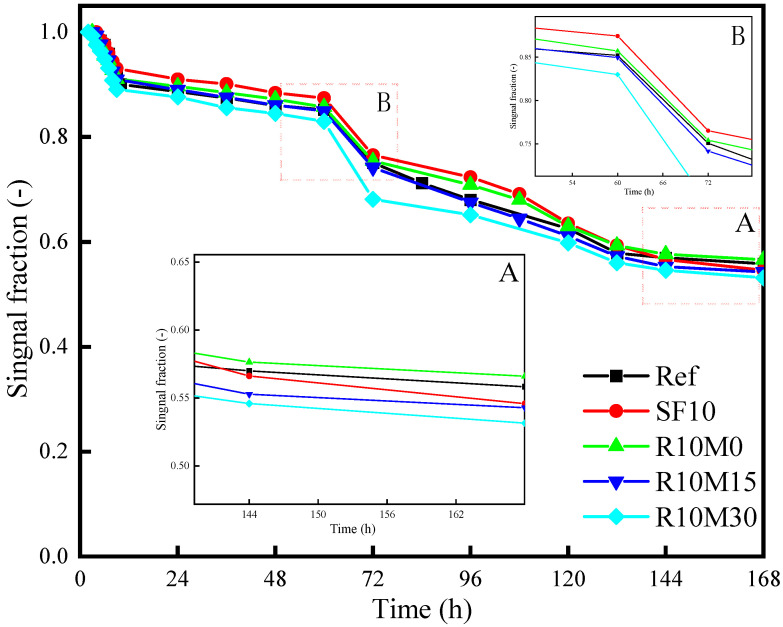
Signal fraction (-) of the free water peak in AAS pastes.

**Figure 11 materials-16-03148-f011:**
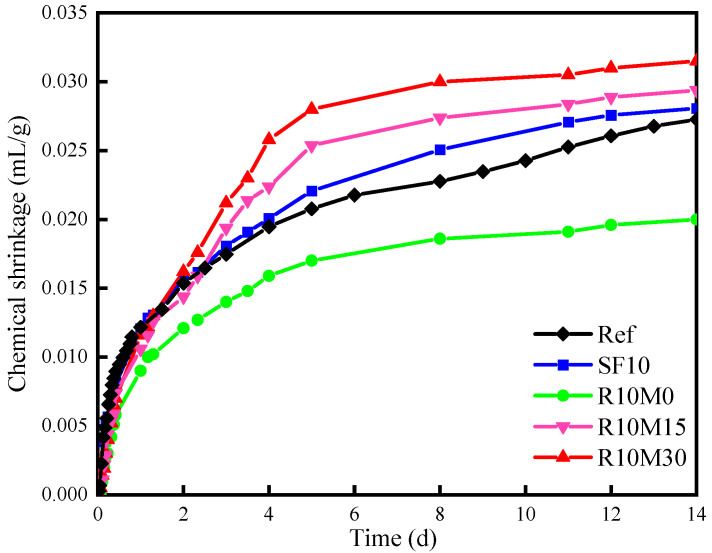
Chemical shrinkage of AAS pastes.

**Figure 12 materials-16-03148-f012:**
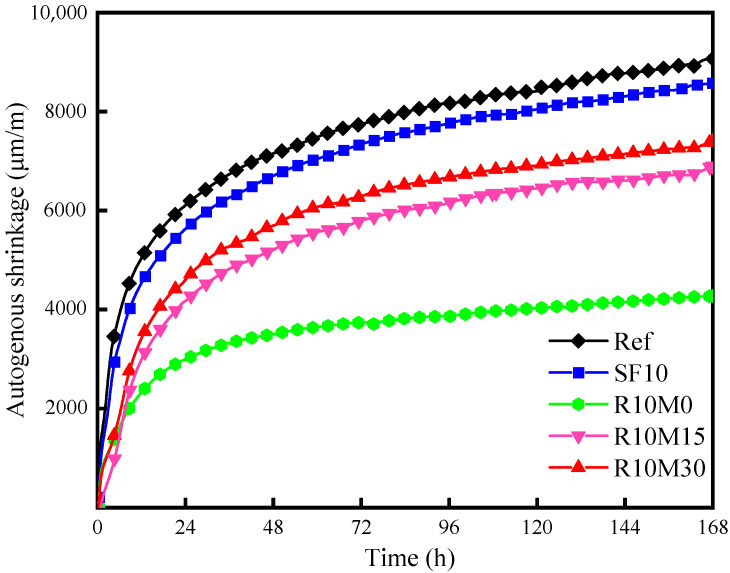
Autogenous shrinkage of AAS pastes.

**Figure 13 materials-16-03148-f013:**
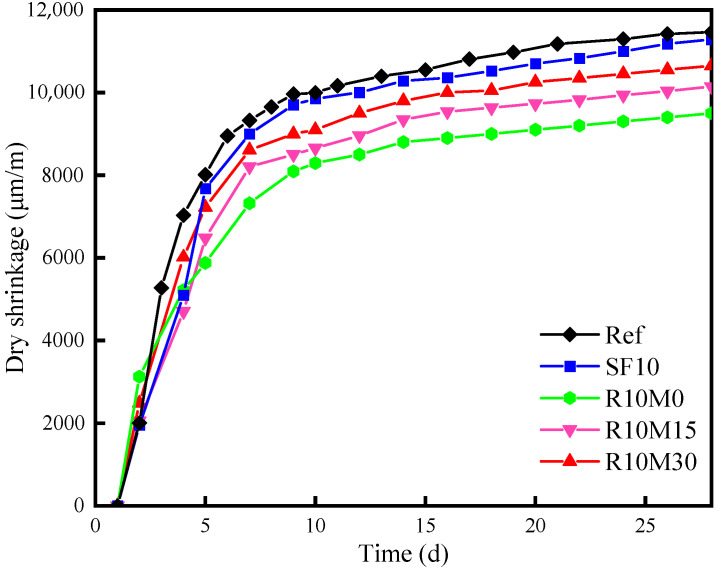
Drying shrinkage of AAS pastes.

**Figure 14 materials-16-03148-f014:**
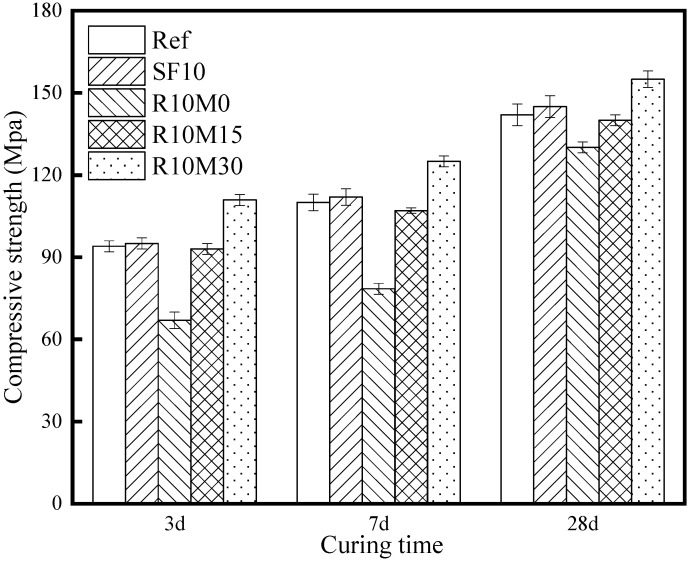
Compressive strength of AAS pastes specimens.

**Table 1 materials-16-03148-t001:** Chemical compositions of materials (wt.%).

Materials	CaO	SiO_2_	Al_2_O_3_	Na_2_O	K_2_O	MgO	SO_3_	P_2_O_5_	Fe_2_O_3_	LOI
Slag	32.63	38.96	15.46	0.30	0.40	7.50	2.55	-	0.34	1.29
SF	0.03	96.80	0.15	-	-	-	-	0.03	0.2	0.79
RHA	0.81	94.63	0.09	0.06	1.33	0.50	0.40	0.60	-	1.3

- not detected.

**Table 2 materials-16-03148-t002:** Mix proportions of samples.

Sample No.	Slag/g	Water Glass/g	NaOH/g	NSP/g	SF/g	RHA/g	W/B
Ref	450	183.6	26.6	4.5	0	0	0.5
SF10	405	183.6	26.6	4.5	45	0	0.5
R10M0	405	183.6	26.6	4.5	0	45	0.5
R10M15	405	183.6	26.6	4.5	0	45	0.5
R10M30	405	183.6	26.6	4.5	0	45	0.5

**Table 3 materials-16-03148-t003:** The deconvolution results.

Sample	Q^0^ (%)	Q^1^-a (%)	Q^1^-b (%)	Q^2^(1Al) (%)	Q^2^-b (%)	Q^2^-p (%)	Q^4^ (%)	Rs (%)
Ref	39.36	9.98	5.11	20.82	20.50	4.23	0	60.64
SF10	29.99	4.92	17.39	14.55	26.88	4.16	2.11	67.90
R10M0	44.64	8.3	5.36	8.22	15.18	18.3	0	55.36
R10M15	30.69	7.53	7.59	14.87	23.78	15.55	0	69.31
R10M30	30.03	8.07	15.92	18.49	18.97	8.54	0	69.97

## Data Availability

Data will be made available on request.

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
