# Peer review of "Effect of Rice Husk Ash on the Properties of Alkali-Activated Slag Pastes: Shrinkage, Hydration and Mechanical Property"

_materials, 2023, doi:10.3390/ma16083148_

Round 1

Reviewer 1 Report

The manuscript “ Effect of rise husk ash on the properties of alkali-activated slag pastes: shrinkage, hydration and mechanical property” deals with influence of rice husk ash on performance alkali activated slag with focus on hydration and shrinkage. The topic is interesting and there is useful information in the article. The analysis follows each other. Sometimes, I miss the explanation of the results.

Before publishing, I would recommend the following points to the attention of the authors:

·         Line 72: State the producers of slag and silica fume

·         Lines 177 – 179: Explain how the addition of RHA reduces the fluidity of pastes

·         Lines 187 – 188: Explain why the incorporation of SF and RHA prolongs the setting time

·         Line 444: What does it mean by thermogravimetric analyses?

Author Response

In the attachment, I have replied to the reviewer's comments point-by-point.

Reviewer 2 Report

Comment on:

Effect of rice husk ash on the properties of alkali-activated slag 2 pastes: shrinkage, hydration and mechanical property 3

By: Xiangguo Li et al.

The paper gives some useful informations on the application of rice husk ash to improve some properties of alkali activated slags. Perhaps the authors should notify that with the technology change to direct reduction of iron ore with H2 (which is the plan in western countries) the slags will not be available to that degree. So, the importance of the informations will decrease in future.

Specific commenst:

Line 32: the application of alkali activated slags is much older than the references suggest.

Line 85: (Na2O = 8.3 Wt. %, SiO2 = 26.6 wt. %)

Table 1: the contents do not sum up to 100 %. Where is the FeO content?

Fig. 2: More than 15 min of milling has not much effect. What is “proper” milling?

Table 2: NSP? Is it a liquid? How high is its water content?

Line 117: Please give exact reference of test procedures.

Lines 123, 127,133: Please tell what the different types of shrinkage mean and what the tell the reader.

Line 150: Where are the BET resultsß

Line 153: Really 50 °C, not K?

Line 154; Barret?

Line 158: Germany

Line 169: Please specify testing machine.

Line 223: The peak does not indicate amorphicity, but more probably cryptocrystallinity.

Table 3: Two digits are too much, the accuracy is not that high. Also, with other numbers in the text.

Line 276: How was the pore structure analyzed. It looks like Hg porosimetry, but this is not decribed in the experimental section.

Line 277: Pore structure after 7 d: Ho was the reaction interrupted and the sample dried?

Line 324: Silicon species have to diffuse out of the pores. That may take some time.

Line 350: internal specific surface area: That has to be discussed. BET gives specific surface area (where are the results?) Hg porosimetry will provide informations of the surface area within zhe pores. Both data are valuable and would help the interpretation. The authors should not discuss fineness but mean particle size and specific surface areas, both overall and internal.

Line 364: accuracy.

Line 375: internal specific surface area: Where are the data?

Line 382: Capillary pressure depends mainly on the diameter of the capillaries.

Line 493: fluidity loss is induced by water uptake of internal pores.

The English spelling and the grammar needs some minor improvements.

Overall: the paper can be accepted after minor to major revisions.

Author Response

1

Reviewer 3 Report

The paper is adequate to publish in the Materials journal, but the authors should take into account the next comments.

-Lines84-85. The chemical composition of the waterglass is wrong. Please, indicate the molarity of NaOH used.

-Lines 98-99. Indicate the meaning of silicate modulus.

-2.3.1. Explain better how was measured the setting time.

-2.3.2. Explain better how was measured the chemical shrinkage and please, explain in detail what are the differences between the three types of shrinkage.

-2.3.3. Please, indicate the samples in their lateral faces have edge-studs preinserted to measure the length modifications.

-2.3.6.What is the instrument employed to carry out the XRD measurements?

-2.3.8. What is the external standard used to do the NMR measurements? And the test conditions? Information about the software used to do the fitting.

-Modify 2.3.9 instead of 2.3.9.1

-2.3.10. What is the instrument employed to determine the mechanical strength? What are the test conditions?

-3.3.1.Lines 197-198. Indicate the position of the maximum or the range of both exothermic peaks.

Line 215 and 424. R-MO???

3.3.2. Indicate the diffraction peaks of the calcite, lines 225-226, and the number of the crystal structures used to identify each phase.

3.3.3. The fit of peaks of the AAS pastes could be not adequate in my humble opinion. When a signal is included in another signal, the authors are considering the same area twice. For example, in Fig. 7(c), Q1-b is included in Q1-a  and in Fig. 7(f) Q1-a is included in Q0. I think that the fit could be improved if the authors consider the same width of the peak for all the components and moreover, with this information about the width of the signal could know if the gel is more or less ordered. This information could also include in Table 3.

3.3.4. The pore structure information should appear before the 29Si MAS NMR information according to experimental section.

Line 291: RHA-O???

Modify 3.3.5 instead of 3.3.5.1.

T2 or T2???

Lines 306-307. Explain how is the free water signal convert to into a percentage of the initial signal strength.

Lines 310-311. This sentence should be rewritten with respect to which R10M30 is the lowest?

-Discussion. The author should include a paragraph to discuss the effect of the RHA presence on the mechanical development.

-References. Full or abbreviated name of journals?

Author Response

In the attachment, I have replied to the reviewer's comments point-by-point

Round 2

Reviewer 1 Report

The manuscript can be published in present form.

Reviewer 2 Report

one final comment on calorimetry (line 234): The early calorimetric peak is sometimes attributed to adsorption of the aqueous liqid to the surfce of the solids.

Reviewer 3 Report

The manuscript is ready for its publication.